# Increased Expression of miR-223-3p and miR-375-3p and Anti-Inflammatory Activity in HDL of Newly Diagnosed Women in Advanced Stages of Breast Cancer

**DOI:** 10.3390/ijms241612762

**Published:** 2023-08-14

**Authors:** Monique de Fatima Mello Santana, Maria Isabela Bloise Alves Caldas Sawada, Aritania Sousa Santos, Mozania Reis, Jacira Xavier, Maria Lúcia Côrrea-Giannella, Andrea Harumy de Lima Hirata, Luiz Henrique Gebrim, Francisco Garcia Soriano, Cleber Pinto Camacho, Marisa Passarelli

**Affiliations:** 1Laboratório de Lípides (LIM 10), Hospital das Clínicas (HCFMUSP), Faculdade de Medicina, Universidade de São Paulo, São Paulo 01246-000, Brazil; moniquemello4@usp.br; 2Programa de Pós-Graduação em Medicina, Universidade Nove de Julho (UNINOVE), São Paulo 01525-000, Brazil; isabelacaldas028@gmail.com (M.I.B.A.C.S.); moszania@gmail.com (M.R.); xavierjacira@gmail.com (J.X.); camachocp@gmail.com (C.P.C.); 3Hospital da Força Aérea de São Paulo, São Paulo 02012-021, Brazil; 4Laboratório de Carboidratos e Radioimunoensaio (LIM 18), Hospital das Clínicas (HCFMUSP), Faculdade de Medicina, Universidade de São Paulo, São Paulo 01246-000, Brazil; aritania@alumni.usp.br (A.S.S.); maria.giannella@fm.usp.br (M.L.C.-G.); andreaharumy@gmail.com (A.H.d.L.H.); 5Unidade Básica de Saúde Dra. Ilza Weltman Hutzler, São Paulo 02472-180, Brazil; 6Centro de Referência da Saúde da Mulher–Hospital Pérola Byington, São Paulo 01215-000, Brazil; lgebrim@uol.com.br; 7Laboratório de Emergências Clínicas (LIM 51), Hospital das Clínicas (HCFMUSP), Faculdade de Medicina, Universidade de São Paulo, São Paulo 01246-000, Brazil; gsoriano@usp.br

**Keywords:** HDL, plasma lipids, breast cancer, microRNA, inflammation

## Abstract

The expression of inflammation-related miRs bound to high-density lipoproteins (HDLs), the anti-inflammatory activity of HDLs isolated from individuals with breast cancer, and controls were determined. Forty newly diagnosed women with breast cancer naïve of treatment and 10 control participants were included. Cholesterol-loaded bone-marrow-derived macrophages were incubated with HDL from both groups and challenged with lipopolysaccharide (LPS). Interleukin 6 (IL6) and tumor necrosis factor (TNF) in the medium were quantified. The miRs in HDLs were determined by RT-qPCR. Age, body mass index, menopausal status, plasma lipids, and HDL composition were similar between groups. The ability of HDL to inhibit IL6 and TNF production was higher in breast cancer compared to controls, especially in advanced stages of the disease. The miR-223-3p and 375-3p were higher in the HDLs of breast cancer independent of the histological type of the tumor and had a high discriminatory power between breast cancer and controls. The miR-375-3p was greater in the advanced stages of the disease and was inversely correlated with the secretion of inflammatory cytokines. Inflammation-related miRs and the anti-inflammatory role of HDLs may have a significant impact on breast cancer pathophysiology.

## 1. Introduction

High-density lipoproteins (HDLs) are heterogenous particles varying from 7.5 to 10.5 nm in diameter, composed of lipids [(free and esterified cholesterol (20%), phospholipids (15%), and triglycerides (5%)] and apolipoproteins (50%, mostly apoA-I, followed by apoA-II, apo AIV, and others) [1]. HDLs encompass a broad range of particles that have been extensively investigated for their role in preventing cardiovascular disease (CVD). Nascent HDL (pre-beta HDL) is produced by the liver and intestine, but predominantly through the activity of lipoprotein lipase during the hydrolysis of triglyceride-enriched lipoproteins in the bloodstream. Pre-beta HDL acts as an excellent acceptor of excess cell cholesterol via ATP-binding cassette transporter A1 (ABCA-1) and as the precursor to mature and larger forms of HDLs, namely HDL_3_, and HDL_2_. The functionality of these mature particles is also linked to their ability to mediate reverse cholesterol transport, removing cell cholesterol and oxysterols through the ATP-binding cassette transporter G1 (ABCG-1). Finally, HDL_2_ delivers cholesterol to the liver allowing for its secretion into bile and excretion in feces. Moreover, HDL subfractions inhibit oxidation, inflammation, and platelet aggregation, and enhance vasodilation, pancreatic insulin secretion, and insulin sensitivity in peripheral organs. The functionality of HDLs is determined by their composition and chemical alterations but also by the proteome and lipidomics of these lipoproteins. Considering all these activities, HDLs have been implicated not only in CVD but also in the development and evolution of other chronic diseases, including breast cancer [2,3]. 

Breast cancer is one of the leading causes of cancer-related death in women [4], and exploring risk factors and modulators helps in the prevention and management of the disease. As a heterogenous disease, breast cancer classification according to the molecular type of tumor is based on the immunohistochemical expression of estrogen and progesterone receptors together with the ki67 index [(luminal A (LA) and luminal B (LB)], the expression of human epidermal growth factor receptor 2 (HER2), or the absence of these receptors, known as triple-negative (TN). This classification serves as the basis for therapeutic choices and clinical outcomes prediction [5,6]. Triple-negative is further classified as basal-like (BL1 and BL2), claudin-low, mesenchymal, luminal androgen receptor, and immunomodulatory.

Plasma lipids are considered modulators of breast cancer development and progression and present a particular profile in TN tumors, possibly helping to drive more lipids for tumor progression [7]. Elevated plasma levels of HDL cholesterol (HDLc) seem to have a protective role against breast cancer [8,9,10,11], although there are still controversial data, and a positive [12,13,14], modest [15], or even lack of association [3,16,17] has also been demonstrated between HDLc and breast cancer development and outcomes. It is possible to consider that in resemblance to CVD, classical metrics of HDLs, including plasma HDLc and apolipoprotein A-I, do not reflect the functional properties of HDLs in breast cancer that may have the contribution of other components of HDLs including bioactive lipids, antioxidant enzymes, and microRNAs (miRs) [1]. By removing excess cell cholesterol, HDLs limit the bioavailability of sterols necessary for cell replication and tumor metastasis. In addition, HDLs reduce oxidative and inflammatory stress modulating the tumor microenvironment.

MicroRNAs (miRs) are highly conserved small, single-stranded, non-coding RNA capable of interacting with target mRNA, impairing translation and protein expression. In circulation, miRs are transported in association with ribonucleoproteins (argonaute, nucleophosmin), exosomes, vesicles, and HDLs [18]. The transportation of miRs carried by HDL favors stability and delivery to recipient cells through the scavenger receptor class B type 1 (SR-B1) regulating several physiological and pathological processes, such as lipid metabolism, inflammation, angiogenesis, and apoptosis [19,20]. The identification of miRs associated with breast cancer is providing a new perspective on the heterogeneity of the disease and has the potential to contribute to the development of novel treatment strategies. MicroRNAs over or under-expressed in the circulation and interstitial fluid are demonstrated in breast cancer as having a regulatory role in multiple pathways related to cell proliferation and the cell cycle [21], although there are no data on the HDL-bounded miRs. In HDL particles isolated from women newly diagnosed with breast cancer and control subjects, we investigated three aspects: (1) the composition in lipids and apoA-I; (2) the differential expression of miRs related to inflammation; and (3) the ability to inhibit the inflammatory response in macrophages. Among the selected 10 miRs related to inflammation in breast cancer and HDLs, a significant and discriminatory upregulation of miR-223-3p and 375-3p in the HDLs of breast cancer women compared to controls was observed, irrespective of the histological type and clinical stage of the disease. Particularly, the expression of miR-375-3p exhibited a strong and positive correlation with HDLs’ capability to inhibit the production of inflammatory cytokines, indicating the potential role of HDL-bound miRs in breast cancer progression and outcomes.

## 2. Results

Anthropometric and clinical data of control and breast cancer groups are depicted in Table 1. The breast cancer group was similar to the control regarding age, body mass index (BMI), and menopausal status. In addition, groups presented similar levels of plasma total cholesterol (TC), triglycerides (TG), apolipoprotein B (apoB), HDLc, LDLc, non-HDLc, and lipid ratios (CT/apoB and TG/HDLc) indicative of small dense LDL formation. No differences were found in these clinical and biochemical parameters when the breast cancer group was subdivided according to the molecular classification of the tumor—LA, LB, HER2, and TN. The composition of isolated HDLs in lipids and apoA-I was similar between the control and breast cancer cases (Table 2). 

HDL particles were analyzed for their ability to inhibit the secretion of inflammatory cytokines by macrophages challenged by lipopolysaccharide (LPS). As shown in Figure 1A,B, HDLs isolated from breast cancer women inhibited 47% and 34% of the secretion of, respectively, IL6 and TNF, as compared to HDLs from the control group. HDLs from breast cancer cases in stage IV presented a great ability to reduce IL6 production as compared to controls and other stages of the disease (Figure 1C). When clinical stages were grouped, a lower secretion of IL6 in advanced stages (III and IV) was observed as compared to early stages (I and II) and controls (Figure 1D). Similar results were observed for TNF with the stage IV individually analyzed, presenting a greater reduction in inflammatory cytokine secretion in comparison to controls and early stages of breast cancer (Figure 1E,F). 

Among the 10 miRs selected by the miRDB, only four were found in association with the HDLs of controls and breast cancer subjects (miR-17-5p, miR-138-1-3p; miR-223-3p, and miR-375-3p; Figure 2A–D) and only miR-223-3p and miR-375-3p were differently expressed in HDLs of breast cancer cases in comparison to the controls (Figure 2C,D). Both miR-223-3p and miR-375-3p had a powerful discriminatory capacity between controls and breast cancer groups according to the ROC curves (Figure 2E,F). 

The enhanced expression of miR-223-3p was observed in all stages of the disease individually analyzed in comparison to the control group and when stages I, II, III, and IV were grouped (Figure 3A,B). On the other hand, the expression of miR-375-3p was only increased in stages III and IV individually compared with the control group. Grouped stages III and IV presented a higher expression of HDL-bound miR-375-3p in comparison to controls and stages I and II grouped (Figure 3C,D). Interestingly, the expression of both miR-223-3p and mir-375-3p was independent of the molecular classification of the tumor being similar in controls, luminal types (A and B), HER2, and TN (Figure 3E,F). 

The expression of miR-223-3p was not associated with the secretion of inflammatory cytokines (Figure 4A,B). On the other hand, the expression of miR-375-3p in HDLs was inversely correlated with the secretion of both IL6 and TNF (Figure 4C,D). 

After exploratory analysis using in-silico strategies, the target genes of miR-223-3p and miR-375-3p were determined using the miRDB platform. Associations were assessed between inflammatory genes (C0021368), mammary inflammation (C0024894), and genes associated with HDLs (C0392885) according to the DisGeNET 7.0 platform [22], which were visualized through a Venn diagram constructed using the interactivenn.net website [23]. For miR-223-3p, the analysis between the groups revealed an association with the *ICAM1* gene, which encodes for intercellular adhesion molecule 1 (ICAM1), and the *TP53* gene, which encodes for the tumor suppressor gene, p53 protein (Figure 5A). As for miR-375-3p, it showed an association with the *JAK2* gene, which encodes for Janus kinase 2 (Figure 5B).

## 3. Discussion

The contribution of HDLs to breast cancer development and outcomes remains a subject of controversy. This is because the conventional metrics used to assess HDL plasma levels may not accurately reflect the diverse composition and functionality of HDL particles in different tissues. Additionally, breast cancer is a heterogeneous disease with varied pathophysiological mechanisms and factors that influence the outcomes. HDL-bound miRs, differentially expressed in breast cancer, could play a significant role in tumor initiation and progression. In this study, a distinct pattern of miR expression was observed, specifically miR-223-3p and miR-375-3p increased in breast cancer cases. Notably, the higher expression of miR-375-3p was positively associated with the HDL’s ability to inhibit the secretion of inflammatory cytokines by LPS-challenged macrophages. Furthermore, the anti-inflammatory activity of HDLs was more prominent in the advanced stages of breast cancer in comparison to early stages and to controls.

The removal of cell cholesterol and oxysterols appears to play a detrimental role in cell proliferation and metastasis. Recent research has demonstrated that HDLs isolated from individuals with advanced-stage breast cancer exhibit a reduced ability to remove cell cholesterol, potentially compromising lipid homeostasis and favoring tumor development. In this particular case, the majority of advanced-stage breast cancer cases were of the TN molecular type. Furthermore, TN cases exhibited higher levels of plasma lipids, which facilitated the channeling of metabolites necessary for cell replication [24]. On the other hand, HDLs in TN breast cancer demonstrated an enhanced ability to delay the oxidation of LDL in vitro compared to HDLs from control subjects, even without alterations in HDL particle composition [25].

In the current investigation, HDL isolated from women with breast cancer exhibited a higher capacity to inhibit the secretion of inflammatory cytokines by LPS-treated macrophages. Remarkably, this property of HDL was consistently demonstrated even after HDLs were removed from the cell culture medium. In other words, HDLs are capable of eliciting a previous protective status in macrophages, thereby preventing an excessive inflammatory response triggered by a potent inflammatory agent. Interestingly, in breast cancer, the anti-inflammatory capacity of HDL was evident across all molecular types of the disease but was more pronounced in the advanced stages of the disease. 

The use of monoclonal antibodies that target the programmed cell death-1 receptor (PD-1) and its ligand PD-L1, known as immune checkpoint inhibitors, has been extensively validated as a highly effective treatment for various types of cancer. By inhibiting the PD-L1/PD-1 axis, these antibodies enhance the immune response against tumor cells. In the case of metastatic TN breast cancer, a favorable response to PD-1 or PD-L1 blockade with pembrolizumab or atezolizumab has been demonstrated [26]. In this sense, considering the increased antioxidant and anti-inflammatory roles of HDLs in breast cancer, it is possible that HDLs, by mitigating oxidative and inflammatory stress, might contribute to a worse prognosis for breast cancer rather than improving tumor progression. As of now, it is not possible to confirm this possibility with the design of the present investigation. A long-term follow-up of the subjects included in the present study will provide us with stronger evidence regarding the role of HDLs in breast cancer survival. 

It is also possible to consider that HDLs are modified by tumor cells or by the tumor microenvironment leading to a dissociation of HDLs as predictors of tumor risk. In this sense, HDLs would function more as markers rather than as determinant factors for tumor development and prognosis. It is possible that the miRs in HDLs represent what was produced by the tumor and the modulation of signaling pathways associated with them.

In addition to the components traditionally associated with the lipoprotein structure, HDLs carry various other substances. Reconstituted HDL has been employed as a delivery vehicle for treating TN breast cancer [27]. Moreover, miRs delivered by HDLs to target cells have been shown to have crucial roles in the pathophysiology of various diseases. Solid tumors, including breast cancer, exhibit an elevated expression of SR-B1, an HDL receptor responsible for selectively removing esterified cholesterol and miRs, and is associated with the poor prognosis of breast cancer [20,28]. Therefore, the delivery of miRs associated with HDLs may contribute to modulating tumor growth. The differential expression of miRs in HDLs could help to explain how HDLs prevent inflammation in breast cancer. 

Tumor cells produce various miRs, and specifically for breast cancer, several of them have already been identified as contributors to tumor differentiation, proliferation, epithelial–mesenchymal transition, invasion, reprogramming, and metastasis. The action of miRs occurs in the tumor microenvironment, as well as between tumor and non-tumor cells. MicroRNAs secreted in the tumor interstitial fluid have been detected and classified into families, serving as functional validation in tumor tissue through the analysis of transcriptome alteration and signaling pathways, particularly in TN tumors [21]. 

Particularly miR-375-3p expression in HDLs was positively correlated with the anti-inflammatory role of HDLs. In this initial exploratory study, we assessed the capacity of HDLs to inhibit inflammation in cholesterol-overloaded macrophages, rather than specifically focusing on tumor cells. Nevertheless, macrophages play a role in promoting tumor development by contributing to the creation of an inflammatory microenvironment and facilitating cross-talk between cells that regulate tumor growth [29].

The increased expression of miR-375-3p in estrogen-receptor-positive breast tumors has been associated with a higher tumor proliferative profile, based on the action of this miR on the target gene *RAS*, dexamethasone-induced 1 (RASD1), which encodes a member of the small GTPase superfamily induced by dexamethasone [30]. Chekhun et al. (2020) did not find any alteration in the expression of miR-375 in the plasma of women with stage II and III breast cancer. However, its content was higher in patients with LA compared to LB tumors [31,32].

Fabris et al. (2016) demonstrated that miR-223 targets epidermal growth factor (EGF), leading to a decrease in EGF signaling, which is crucial for normal mammary gland development and breast cancer. A study conducted on individuals before and after surgery and intraoperative radiotherapy (IORT) showed a positive regulation of miR-223 expression in the mammary gland after IORT, resulting in local EGF release and a decreased survival of cancer cells. As a response to this, a reduction in circulating miR-223 concentration was observed after surgery [33]. 

Yoshikawa et al. (2018) evaluated circulating exosomes from 185 patients with invasive ductal carcinoma, revealing a higher expression of miR-223-3p compared to the control group in the microarray analysis. Moreover, in MCF-7 cells transfected with miR-223-3p, an increase in proliferation and invasive capacity was observed [34]. On the other hand, Citron et al. (2020) demonstrated that miR-223 expression is negatively regulated in different molecular types of breast cancer, particularly in luminal and HER2-positive types. When evaluating breast cancer cells (MCF-7, SJBR3, MDA-MB-231, and MDA+MB-435), it was observed that miR-223 expression was downregulated in all cell lines compared to control cells [35].

In vitro studies using inflammatory breast cancer cell lines have demonstrated that HDL enhances cellular sensitivity to radiation, primarily through the modulation of intracellular cholesterol levels. However, this effect is counteracted by the overexpression of miR-33, which reduces the expression of ABCA-1. Interestingly, a high expression of miR-33a, known to decrease HDL levels, has been associated with shorter overall survival in breast cancer patients undergoing radiotherapy [36].

The in-silico analysis revealed the association of miR-223-3p with *ICAM* and *TP53* genes. The expression of ICAM is positively regulated in response to a variety of inflammatory mediators [37]. Additionally, ICAM1 shows an increased expression in various types of cancer, being associated with advanced disease stages of the disease, chemotherapy resistance, and lower survival [38]. Guo et al. (2014) demonstrated that ICAM1 is a potential molecular target and biomarker for therapy and diagnosis in women with TN tumors [39]. The *TP53* gene encodes the p53 tumor protein, which acts by regulating cell division and preventing cell proliferation [40]. Mutations in this gene are present in approximately 50% of human cancers, making it the most common target for genetic alterations in the neoplastic process [40], although in breast cancer, the frequency of *TP53* gene mutations is not very high (20–30%). Nevertheless, identifying the type of mutation can lead to a potential therapeutic target and the prediction of survival in these patients [41]. The association analysis for miR-375-3p pointed to the JAK2 gene, which encodes for Janus kinase 2, a tyrosine kinase that is phosphorylated in response to the action of various cytokines [42]. The JAK2 signaling transduction pathway phosphorylates STAT3, triggering the activation of pathways related to the production of chemokines released in the tumor microenvironment, promoting the attraction of macrophages [43]. In untreated women with TN breast cancer, an increase in *JAK2* gene amplification has been reported and linked with chemotherapy resistance [44].

To the best of our knowledge, this is the first demonstration of a distinct pattern of miR bound to HDL in breast cancer, which is correlated with the enhanced ability of HDLs in reducing inflammation, although causality needs to be proven. Given the scarcity of investigations into the molecular mechanisms connecting HDLc and breast cancer, these findings underscore the notion that alterations in HDL functionality, regardless of HDLc and lipoprotein composition, may have implications for the pathophysiology of breast cancer. Further studies are imperative to gain a comprehensive understanding of the HDL contribution to long-term clinical outcomes in breast cancer. 

## 4. Material and Methods

Forty participants were selected from a large group of 201 women newly diagnosed with breast cancer between 18 and 80 years old in any clinical stage and with the molecular classification of the tumor recruited at Hospital Pérola Byington, Sao Paulo, Brazil. The control group consisted of ten women without any type of cancer recruited at the Universidade de São Paulo and at the Unidade Básica de Saúde Dra. Ilza Weltman Hutzler. Women with a previous history of any cancer, in situ breast disease, with diabetes mellitus, chronic kidney disease (estimated glomerular filtration rate <60 mL/min/1.73 m^2^), autoimmune diseases, or who were smokers, alcoholics, or in use of contraceptives, or on hormone replacement therapy or pregnant were not included. Participants were informed about the study and signed an informed written consent previously approved by institutional Ethics Committees, including approval for publication (Universidade Nove de Julho, # 3.139.460; Centro de Referência da Saúde da Mulher, Hospital Pérola Byington, #3.225.220; and Hospital das Clínicas da Faculdade de Medicina da Universidade de São Paulo, #3.317.909), in accordance with the Declaration of Helsinki. 

In the breast cancer group, women were selected according to the molecular classification of tumors obtained from medical records. In accordance with the American College of Pathologists [45,46], tumors were classified as positive for hormone receptors (estrogen and progesterone) in which >1% of the tumor cells showed positive nuclear staining of moderate to strong intensity on immunohistochemistry with Ki67 < 14% (LA; n = 10) or Ki67 > 14% (LB; n = 10). Samples with > 10% of invasive tumor cells with strong staining in the plasma membrane for HER2 were considered HER2 positive (n = 10). In case of moderate staining in >10% of the cells or strong in < 10% of the cells, the sample was re-evaluated by in situ hybridization and was considered positive if it had an HER2/centromere ratio > 2.0; or a HER2/centromere ratio < 2.0 with mean HER2 > 6 signals per cell (greater than 120 signals in 20 nuclei). Tumor samples without the expression of hormone receptors and HER2 were categorized as TN (n = 10). 

### 4.1. Blood Collection

Venous blood was drawn after 12 h of fasting and plasma was immediately isolated after centrifugation (3000 rpm, 4 °C, 15 min). Plasma lipids (TC, TG, HDLc) were determined by enzymatic techniques. HDLc was measured after the precipitation of apoB-containing lipoproteins in plasma treated with dextran sulfate and magnesium chloride. Low-density lipoprotein cholesterol (LDLc) was determined by the Friedewald formula [47]. ApoB was quantified by immunoturbidimetry (Randox Lab. Ltd. Crumlin, UK). Body mass and height were obtained from all subjects.

### 4.2. Isolation of Plasma Lipoproteins

Plasma from all participants was submitted to discontinuous density ultracentrifugation to isolate HDL (D = 1.063–1.21 g/mL). The lipoprotein fraction was immediately frozen at −80 °C in a 5% saccharose solution. HDL composition in lipids (TC, TG, and phospholipids (PL) was determined by enzymatic techniques, and apoA-I by immunoturbidimetry (Randox Lab. Ltd., Crumlin, UK). Low-density lipoprotein (LDL; D = 1.019–1.063 g/mL) was obtained by the sequential ultracentrifugation of plasma from healthy volunteers and was purified by discontinuous density ultracentrifugation. Acetylated LDL was produced by incubation with acetic anhydride as previously described [48] following extensive dialysis against phosphate-buffered saline. The protein concentration was determined using the Lowry technique [49]. 

### 4.3. Determination of the Anti-Inflammatory Activity of HDL from Controls and Breast Cancer Cases

The study was approved by the Institutional Animal Care and Research Advisory Committee (Universidade de Sao Paulo # 1612/2021) according to the U.S. National Institutes of the Health Guide for the Care and Use of Laboratory Animals. C57BL/6 J mice, aged 2–48 weeks, were housed in a conventional animal facility at 22 ± 2 °C under a 12-h light/dark cycle with free access to commercial chow (Nuvilab CR1, São Paulo, Brazil) and drinking water. Animals were euthanized with an intraperitoneal overdose of ketamine hydrochloride (Ketalar) (300 mg/kg of body weight) and xylazine hydrochloride (Rompun) (30 mg/kg of body weight), in accordance with the norms of the National Council for the Control of Animal Experimentation (CONCEA) of the Ministry of Science, Technology, and Innovation (MCTI). Undifferentiated bone-marrow cells were obtained from animals’ tibias and femurs and cells were differentiated into macrophages as previously described [50]. Macrophages were overloaded with acetylated LDL (50 µg/mL) for 24 h, and after washing they were treated with the HDLs (50 µg/mL; 24 h) of controls and breast cancer subjects. Then, cells were challenged with LPS (1 µg/mL) for 24 h. The medium was collected to determine the concentration of inflammatory cytokines, interleukin 6 (IL6), and tumor necrosis factor (TNF) by ELISA (R&D System).

### 4.4. Determination of miR Expression in HDL from Controls and Breast Cancer Cases

Inflammation-related miRs associated with HDL were found using the DisGeNET 7.0 platform [22] and associations were evaluated between inflammatory genes (C0021368), mammary inflammation (C0024894), and genes associated with HDLs (C0392885). These associations were visualized using a Venn diagram constructed through the interactivenn.net website [23] (Figure 6). After an exploratory analysis using an in-silico approach, the target genes of miRs differentially expressed in HDLs were determined by the miRDB (microRNA target prediction database) platform.

The presence of free hemoglobin was determined by measuring the absorbance at 414 nm using a Nanodrop 2000 spectrophotometer (Thermo Scientific, Waltham, MA, USA). All HDL samples exhibited absorbance values ≤ 0.2, thus confirming their suitability for total RNA, including miR, extraction. For total RNA extraction, the miRNeasy Serum/Plasma Kit (Qiagen, Germany) was used following the manufacturer’s instructions. Briefly, 1000 μL of Trizol was added to each tube containing 20 μL of the sample, vigorously homogenized for 10 s, and incubated for 5 min at room temperature. The efficiency of total RNA extraction was monitored by adding 2 μL of synthetic miRNA-39 (*Caenorhabditis elegans*—cel-miR-39) spike-in at a concentration of 2.5 × 105 pmol (Thermo Phosphorylated). The reverse transcription reaction was performed using specific kits (TaqMan^®^ Advanced miRNA cDNA Synthesis Kit; Applied Biosystems—Thermo Fisher Scientific, USA). Real-time quantitative polymerase chain reaction (RT-qPCR) was performed using TaqMan MicroRNA Assays (Applied Biosystems, Foster City, CA, USA). The products of the reverse transcription reaction were pre-amplified following the manufacturer’s protocol. The pre-amplified samples were stored at −20 °C in a freezer for 24 h. For RT-qPCR, probes (A25576) specific to the cDNA sequences of the target miRs were used. The cDNA was diluted 1:10, containing 5 μL of the pre-amplified product and 45 μL of TE buffer. In a 1.5-mL microtube, the PCR reaction mixture was prepared, consisting of 10 μL of TaqMan Fast Advanced Master Mix (2×), 1 μL of TaqMan Advanced miRNA Assay (20×), and 4 μL of nuclease-free water. This mixture (15 μL) was added to each well of the plates (MicroAmp^®^ Fast Optical 96-Well Reaction Plate with barcode, 0.1 mL—Thermo Fisher Scientific, USA), along with 5 μL of the diluted cDNA, resulting in a final volume of 20 μL in each well. The plate was then sealed with adhesive tape (MicroAmp™ Optical Adhesive Film—Thermo Fisher Scientific, USA) and centrifuged at room temperature to ensure even distribution of the samples within each well. The RT-qPCR was performed on the StepOnePlus system with the following cycling conditions: 95 °C for 20 s for 1 cycle, followed by 40 cycles of 95 °C for 1 s and 60 °C for 20 s, with a final step at 4 °C.

### 4.5. Statistical Analysis

Sample normality was analyzed by the Shapiro–Wilk test and the Grubbs test was applied to identify possible outliers. Non-parametric data were represented by the median with lower (25%) and upper (75%) quartiles and compared by the Mann–Whitney or Kruskal–Wallis test. Frequencies were compared using the Chi-square test. GraphPad Prisma (version 5.04) for Windows, Microsoft^®^ Excel for Mac (version 16.52), and IBM^®^ SPSS Statistics (version 27.0) software were used for data tabulation and analysis. A value of *p* < 0.05 was considered statistically significant.

## Figures and Tables

**Figure 1 ijms-24-12762-f001:**
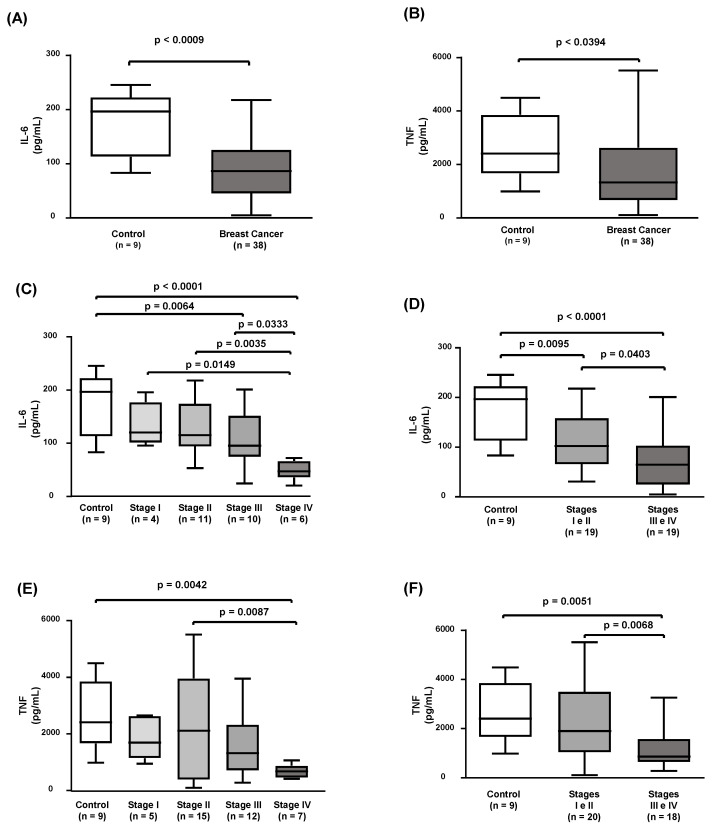
HDL anti-inflammatory activity in controls and breast cancer cases. Cholesterol-overloaded bone-marrow-derived macrophages were incubated with HDL (50 µg/mL) isolated from controls and breast cancer cases for 24 h. After washing, cells were challenged with LPS (1 µg/mL) for 24 h; medium was collected for the measurement of IL6 (**A**) and TNF (**B**) by ELISA. Comparisons were performed using the Mann–Whitney test (controls vs. breast cancer, (**A**,**B**) or the Kruskal–Wallis test for isolated clinical stages (**C**,**E**) or combined early and advanced stages of breast cancer (**D**,**F**).

**Figure 2 ijms-24-12762-f002:**
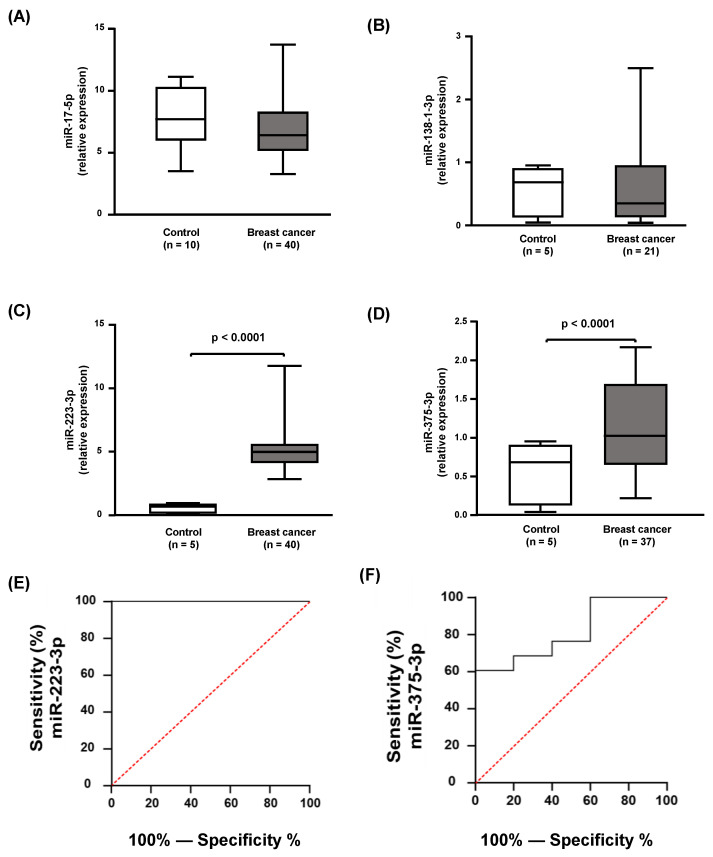
Inflammation-related miRs bound to HDL in controls and breast cancer cases. The expression of miRs associated with HDL isolated from controls and breast cancer cases was determined by RT-qPCR (**A**–**D**). Comparisons were performed using the Mann–Whitney test. The ROC curves (**E**,**F**) showed a higher discriminatory power for miR-233-3p and miR-375-3p between controls and breast cancer groups.

**Figure 3 ijms-24-12762-f003:**
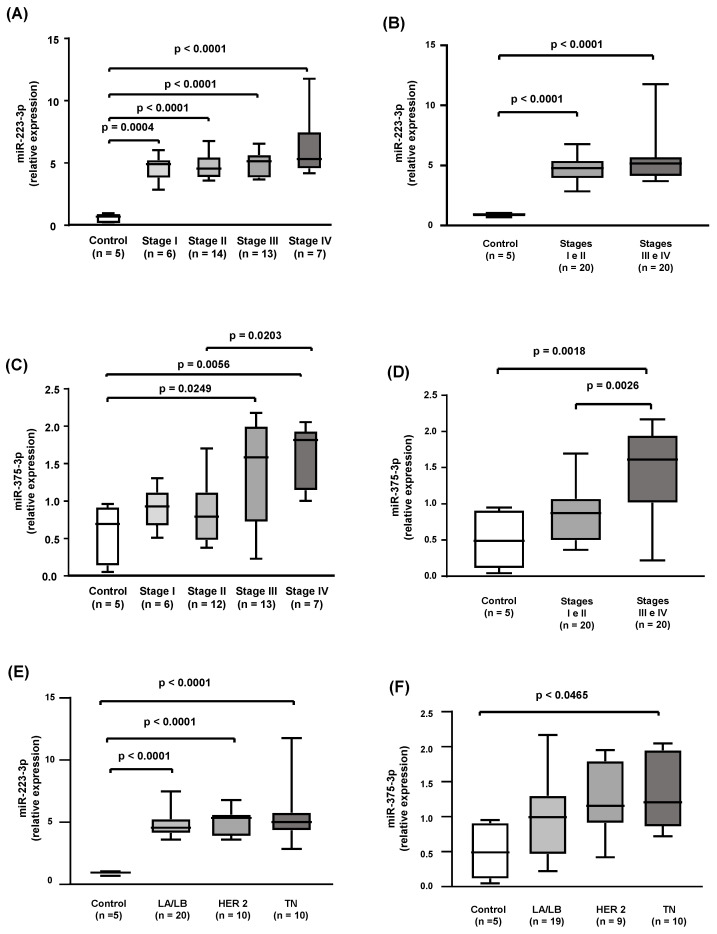
Inflammation-related miRs bound to HDL in controls and breast cancer according to the clinical stage and molecular type of the disease. The expression of miRs associated with HDL isolated from controls and breast cancer cases was determined by RT-qPCR and compared according to the stage of the disease (**A**–**D**) and to the molecular classification of the tumor (**E**,**F**) by the Kruskal-Wallis test.

**Figure 4 ijms-24-12762-f004:**
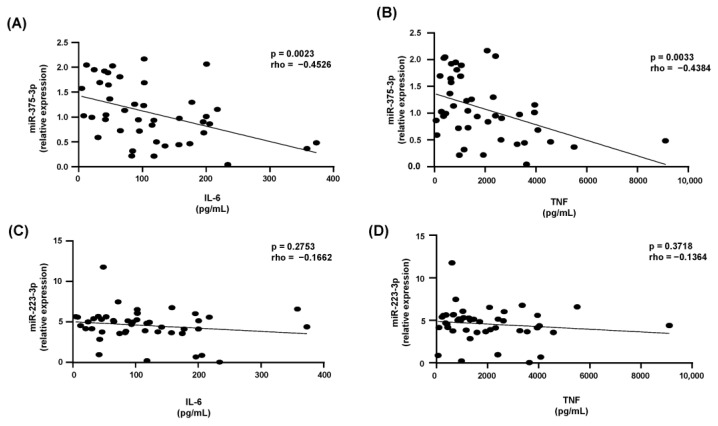
Association between miR-223-3p and miR-375-3p with the secretion of inflammatory cytokines. The expression of miRs associated with HDLs was correlated with the secretion of inflammatory cytokines produced by bone-marrow-derived macrophages challenged by LPS after being incubated with HDLs from controls and breast cancer cases analysis were, performed using the Spearman test. (**A**,**B**): miR-375-3p with IL-6 and TNF, respectively; (**C**,**D**): miR233-3p with IL-6 and TNF, respectively.

**Figure 5 ijms-24-12762-f005:**
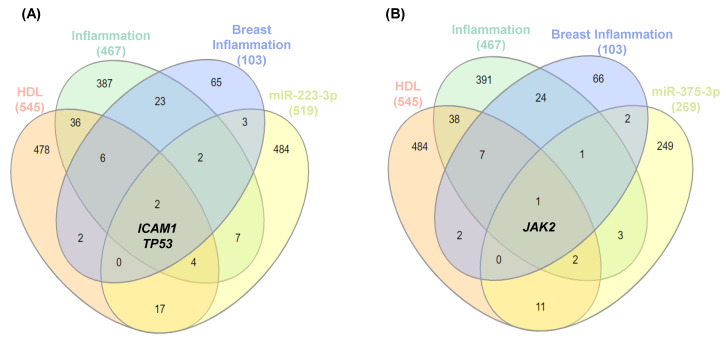
Target genes of miR-223-3p and miR-375-3p bound to HDLs. The target genes of miR-223-3p (**A**) and miR-375-3p (**B**) were determined using the miRDB platform.

**Figure 6 ijms-24-12762-f006:**
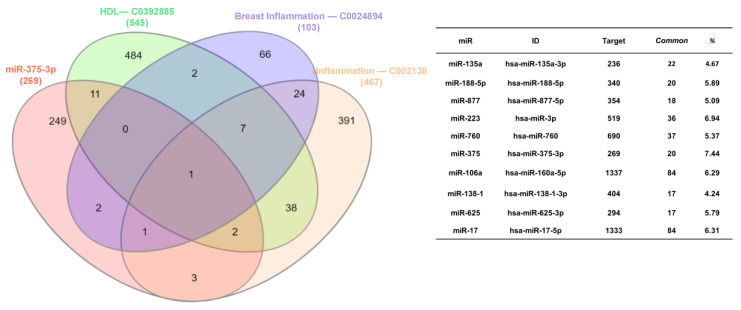
Prediction of inflammation-related miRs bound to HDL. Associations were assessed between inflammatory genes (C0021368), mammary inflammation (C0024894), and genes associated with HDL (C0392885) according to the DisGeNET 7.0 platform, which were visualized through a Venn diagram constructed using the interactivenn.net website.

**Table 1 ijms-24-12762-t001:** Anthropometric and clinical data of controls and breast cancer women.

	Control	Breast Cancer	*p*
n	10	40	
Age (year)	54 (42–60)	53 (45–59)	0.9709
BMI (Kg/m^2^)	28 (25–31)	27 (24–31)	0.6105
Pre-menopause %	4 40	6 60	-
Post-menopause %	19 47.5	21 52.5	-
TC (mg/dL)	172 (158–193)	180 (155–200)	0.7606
TG (mg/dL)	88 (65–110)	99 (79–129)	0.3810
apoB (mg/dL)	135 (117–146)	116 (83–143)	0.1479
HDLc (mg/dL)	41 (31–62)	39 (31–45)	0.4691
LDLc (mg/dL)	109 (102–114)	114 (94–139)	0.5593
TC/apoB	1.2 (1.1–1.4)	1.5 (1.2–1.9)	0.0569
TG/HDLc	1.9 (1.2–3.3)	2.6 (1.5–3.5)	0.3099

BMI = body mass index; TC = total cholesterol; TG = triglycerides; apoB = apolipoprotein B; HDLc = HDL cholesterol; LDLc = LDL cholesterol. Comparisons were performed using the Mann–Whitney test.

**Table 2 ijms-24-12762-t002:** Composition of HDLs isolated from control and breast cancer women.

	Control	Breast Cancer	*p*
n	10	40	
TC (mg/dL)	40 (28–58)	36 (27–46)	0.3892
TG (mg/dL)	15 (11–25)	13 (10–16)	0.1667
PL (mg/dL)	76 (60–89)	84 (71–118)	0.3504
apoA-I (mg/dL)	85 (66–119)	110 (74–127)	0.3381

TC = total cholesterol; TG = triglycerides; PL = phospholipids; apoA-I = apolipoprotein A-I. Comparisons were performed using the Mann–Whitney test.

## Data Availability

All data reported are included in the manuscript and raw data can be kindly shared upon personal request to the corresponding author MP (m.passarelli@fm.usp.br).

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
