# Peer review of "Increased Expression of miR-223-3p and miR-375-3p and Anti-Inflammatory Activity in HDL of Newly Diagnosed Women in Advanced Stages of Breast Cancer"

_ijms, 2023, doi:10.3390/ijms241612762_

Round 1

Reviewer 1 Report

This study represents an interesting investigation into the expression of inflammation-related miRs and the anti-inflammatory activity of HDL in breast cancer. However, there are some issues:

Major points:

1.    Line 78-88: does the final part of the introduction summarize the purpose and results of the study? If so, the authors should indicate it more clearly. 

2.    The number of patients is four times higher than that of controls. To perform a more accurate statistical analysis, the authors should increase the number of healthy subjects.

3.    The number of controls and patients reported in the graphs is different from those indicated in the tables and “Materials and Methods” section. Why is that?

Minor points

1.    The authors should change the colour of the black boxes in the graphs because the median line is not visible.

2.    The correlation coefficient for the Spearman test is represented by "rho" (ρ) and not "r". The authors should correct this information.

3. The authors should correct some editing mistakes.

Author Response

Reviewer 1

This study represents an interesting investigation into the expression of inflammation-related miRs and the anti-inflammatory activity of HDL in breast cancer. However, there are some issues:

Major points:

  1. Line 78-88: does the final part of the introduction summarize the purpose and results of the study? If so, the authors should indicate it more clearly. 

The authors appreciate the reviewer's comments, which assist in improving the manuscript. The sentence that summarizes the study and main conclusion has been improved in the new version of the manuscript (line 89). “In HDL particles isolated from women newly diagnosed with breast cancer and control subjects, we investigated three aspects: 1) the composition in lipids and apoA-I, 2) the differential expression of miRs related to inflammation, and 3) the ability to inhibit the inflammatory response in macrophages. It was observed a significant and discriminatory upregulation of miR-223-3p and 375-3p in the HDL of breast cancer women compared to controls, irrespective of the histological type and clinical stage of the disease. Particularly, the expression of miR-375-3p exhibited a strong and positive correlation with HDL's capability to inhibit the production of inflammatory cytokines, indicating the potential role of HDL-bound miRs in breast cancer progression and outcomes”.

  1. The number of patients is four times higher than that of controls. To perform a more accurate statistical analysis, the authors should increase the number of healthy subjects.

Control subjects (n=10) were carefully matched with breast cancer cases in terms of age and body mass index to prevent these variables from influencing HDL composition and function. Furthermore, the strict exclusion criteria (diabetes mellitus, chronic diseases, previous history of cancer, use of contraceptives or hormone replacement therapy) limited the inclusion of a larger number of individuals in the control group. Breast cancer cases were selected based on the molecular classification of tumors (luminal A, B, HER2, and triple-negative), as these classifications impact the differential pathophysiology of breast cancer and predict distinct clinical outcomes for the disease. Thus, the pairing process was carried out considering the molecular types of breast cancer (n=10 per group). It is important to note that the data obtained from the control group exhibited minimal variability, reflecting the homogeneity of the group.

  1. The number of controls and patients reported in the graphs is different from those indicated in the tables and “Materials and Methods” section. Why is that?

The Grubbs test was applied to identify potential outliers, and subsequently, some subjects were excluded from some analyses. This has been elucidated in line 418 of the manuscript.

Minor points

  1. The authors should change the color of the black boxes in the graphs because the median line is not visible.

The authors fully agree and have changed the color of the figures.

  1. The correlation coefficient for the Spearman test is represented by "rho" (ρ) and not "r". The authors should correct this information.

The authors have corrected the information.

  1. The authors should correct some editing mistakes.

The authors are thankful for the observation. The manuscript was revised and editing mistakes were corrected

Reviewer 2 Report

Dear authors, 

the paper is well written, with interesting findings, 

I hereby present you with observations for improvement of your paper:

- LPS abbreviation is not explained (1st mentioned in line 23, without the abbreviation)

- as it is correctly described, the HDL family is a diverse group and the "classical metrics of HDL including plasma HDLc and apolipoprotein A-I do not reflect the functional properties of HDL"... it would be appropriate to add a few thoughts regarding the different HDL subclasses regarding their function/atherogenicity.

- the selected miR - 223-3-p and 375-3p - are first mentioned in line 82 but without at least a marginal explanation of why you monitored the expression of these miRs, even though a detailed explanation is given in the results section, line 124.

- Figure 1: ...were incubated "with?" HDL...

- Disgenet is mentioned in line 124, later in lines 164-170 explained with a different description, as DisGeNET 7.0 platform (line 173 miRDB platform?)

- line 292: ...demonstration of a distinct pattern of miR bound to HDL in breast cancer, which is correlated with the enhanced ability of HDL in reducing inflammation." it does not follow from the work that miRs are behind the reduced ability of HDL to reduce inflammation, only that they are contained in an altered degree.

-line 379: ". Only HDL samples with absorbance values ≤ 0.2 were considered suitable 379 for total RNA, including miR, extraction. " Were there any samples excluded from the study?

Author Response

Reviewer 2

Dear authors, 

the paper is well written, with interesting findings, I hereby present you with observations for improvement of your paper:

- LPS abbreviation is not explained (1st mentioned in line 23, without the abbreviation)

 The authors appreciate the reviewer's comments, which assist in improving the manuscript. The abbreviation for LPS was correctly indicated, ensuring clarity for readers.

- as it is correctly described, the HDL family is a diverse group and the "classical metrics of HDL including plasma HDLc and apolipoprotein A-I do not reflect the functional properties of HDL"... it would be appropriate to add a few thoughts regarding the different HDL subclasses regarding their function/atherogenicity.

The authors have included more detailed information about HDL subclasses and functionality that contribute to preventing cardiovascular disease and other chronic diseases such as breast cancer (see below).

Line 39: HDL encompasses a broad range of particles that have been extensively investigated for their role in preventing cardiovascular disease (CVD). Nascent HDL (pre-beta HDL) is produced by the liver and intestine, but predominantly through the activity of lipoprotein lipase during the hydrolysis of triglyceride-enriched lipoproteins in the circulation. Pre-beta HDL acts as an excellent acceptor of excess cell cholesterol via ATP-binding cassette transporter A1 (ABCA-1) and as the precursor to mature and larger forms of HDL, namely HDL3, and HDL2. The functionality of these mature particles is also linked to their ability to mediate reverse cholesterol transport, removing cell cholesterol and oxysterols through the ATP-binding cassette transporter G1 (ABCG-1). Finally, HDL2 delivers cholesterol to the liver allowing its secretion into bile and excretion in feces. Moreover, HDL subfractions inhibit oxidation, inflammation, and platelet aggregation, and enhance vasodilation, pancreatic insulin secretion, and insulin sensitivity in peripheral organs. The functionality of HDL is determined by the composition and chemical alterations but also by the proteome and lipidomics of these lipoproteins. Considering all these activities, HDL has been implicated not only in CVD but also in the development and evolution of other chronic diseases, including breast cancer [2,3].

- the selected miR - 223-3-p and 375-3p - are first mentioned in line 82 but without at least a marginal explanation of why you monitored the expression of these miRs, even though a detailed explanation is given in the results section, line 124.

The authors have improved the description pertaining to the measurement of the selected miR-223-3p and 375-3p, as properly suggested. Line 89:  “In HDL particles isolated from women newly diagnosed with breast cancer and control subjects, we investigated three aspects: 1) the composition in lipids and apoA-I, 2) the differential expression of miRs related to inflammation, and 3) the ability to inhibit the inflammatory response in macrophages. Among the selected 10 miRs related to inflammation in breast cancer and HDL, it was observed a significant and discriminatory upregulation of miR-223-3p and 375-3p in the HDL of breast cancer women compared to controls, irrespective of the histological type and clinical stage of the disease. Particularly, the expression of miR-375-3p exhibited a strong and positive correlation with HDL's capability to inhibit the production of inflammatory cytokines, indicating the potential role of HDL-bound miRs in breast cancer progression and outcomes”.

- Figure 1: ...were incubated "with?" HDL...

The authors have corrected the sentence

- Disgenet is mentioned in line 124, later in lines 164-170 explained with a different description, as DisGeNET 7.0 platform (line 173 miRDB platform?)

The authors have corrected the information (line 136)

- line 292: ...demonstration of a distinct pattern of miR bound to HDL in breast cancer, which is correlated with the enhanced ability of HDL in reducing inflammation." it does not follow from the work that miRs are behind the reduced ability of HDL to reduce inflammation, only that they are contained in an altered degree.

The higher anti-inflammatory activity of HDL, as evidenced by the reduced production of inflammatory cytokines by macrophages, showed a positive correlation with the content of miR-375 in these lipoproteins. However, the authors acknowledge that the data obtained do not allow for concluding causality regarding this association. This aspect is now better discussed in line 304 of the revised version of the manuscript, as follows: “To the best of our knowledge, this is the first demonstration of a distinct pattern of miR bound to HDL in breast cancer, which is correlated with the enhanced ability of HDL in reducing inflammation although causality needs to be proven”.

-line 379: ". Only HDL samples with absorbance values ≤ 0.2 were considered suitable for total RNA, including miR, extraction. " Were there any samples excluded from the study?

The authors have corrected the sentence as follows: All HDL samples exhibited absorbance values ≤ 0.2, thus confirming their suitability for total RNA, including miR, extraction. (line 391).

Round 2

Reviewer 1 Report

Dear authors, thank you for the corrections and clarifications added to your manuscript.